# An Overview of *Angiostrongylus cantonensis* (Nematoda: Angiostrongylidae), an Emerging Cause of Human Angiostrongylosis on the Indian Subcontinent

**DOI:** 10.3390/pathogens12060851

**Published:** 2023-06-20

**Authors:** Divakaran Pandian, Tomáš Najer, David Modrý

**Affiliations:** 1Department of Veterinary Sciences, Faculty of Agrobiology, Food and Natural Resources/CINeZ, Czech University of Life Sciences Prague, 16500 Prague-Suchdol, Czech Republic; najer@af.czu.cz; 2Department of Botany and Zoology, Faculty of Science, Masaryk University, 61137 Brno, Czech Republic; 3Biology Center, Institute of Parasitology, Czech Academy of Sciences, 37005 Ceske Budejovice, Czech Republic

**Keywords:** *Angiostrongylus cantonensis*, human angiostrongyliasis, Indian subcontinent, eosinophilic meningitis

## Abstract

Human angiostrongylosis is an emerging zoonosis caused by the larvae of three species of metastrongyloid nematodes of the genus *Angiostrongylus,* with *Angiostrongylus cantonensis* (Chen, 1935) being dominant across the world. Its obligatory heteroxenous life cycle includes rats as definitive hosts, mollusks as intermediate hosts, and amphibians and reptiles as paratenic hosts. In humans, the infection manifests as *Angiostrongylus* eosinophilic meningitis (AEM) or ocular form. Since there is no comprehensive study on the disease in the Indian subcontinent, our study aims at the growing incidence of angiostrongylosis in humans, alongside its clinical course and possible causes. A systematic literature search revealed 28 reports of 45 human cases from 1966 to 2022; eosinophilic meningitis accounted for 33 cases (75.5%), 12 cases were reported as ocular, 1 case was combined, and 1 case was unspecified. The presumed source of infection was reported in 5 cases only. Importantly, 22 AEM patients reported a history of eating raw monitor lizard (*Varanus* spp.) tissues in the past. As apex predators, monitor lizards accumulate high numbers of L3 responsible for acute illness in humans. For ocular cases, the source was not identified. Most cases were diagnosed based on nematode findings and clinical pathology (primarily eosinophilia in the cerebrospinal fluid). Only two cases were confirmed to be *A. cantonensis*, one by immunoblot and the other by q-PCR. Cases of angiostrongylosis have been reported in Delhi, Karnataka, Kerala, Maharashtra, Madhya Pradesh, Puducherry, Telangana, and West Bengal. With a population of more than 1.4 billion, India is one of the least studied areas for *A. cantonensis*. It is likely that many cases remain undetected/unreported. Since most cases have been reported from the state of Kerala, further research may focus on this region. Gastropods, amphibians, and reptiles are commonly consumed in India; however, typical preparation methods involve cooking, which kills the nematode larvae. In addition to studying rodent and mollusk hosts, monitor lizards can be used as effective sentinels. Sequence data are urgently needed to answer the question of the identity of *Angiostrongylus*-like metastrongylid nematodes isolated from all types of hosts. DNA-based diagnostic methods such as q-PCR and LAMP should be included in clinical diagnosis of suspected cases and in studies of genetic diversity and species identity of nematodes tentatively identified as *A. cantonensis*.

## 1. Introduction

*Angiostrongylus* eosinophilic meningitis caused by *Angiostrongylus cantonensis* is a zoonotic disease that is currently spreading in the tropics and subtropics, with sporadic cases also occurring in temperate climatic zones [1,2]. The disease is caused by a metastrongyloid nematode, the rat lungworm, *Angiostrongylus cantonensis*. In humans, larval migration of this nematode typically results in eosinophilic meningitis and other central nervous system disorders. The increasing incidence and frequency of angiostrongylosis outbreaks has led to the disease being classified as an emerging infectious disease in Southeast Asian countries and in some invaded areas such as Australia and islands in the Pacific region.

*Angiostrongylus cantonensis* was first discovered in China (Chen, 1935), and it is primarily in Southeast Asia that is holding a tradition of its research in the Old World [2,3], reflecting the growing incidence of the disease [4]. Recently, infections of humans by the closely related *Angiostrongylus malaysiensis* (Bhaibulaya & Cross, 1971) have been reported in Southeast Asia [5], and the precise differentiation of the two nematode species deserves further attention, as does the possible neurotropism (the ability of larvae to specifically invade the brain and central nervous system) of other members of the genus.

The Indian subcontinent has never been systematically studied and reviewed for the presence of *A. cantonensis*. There, angiostrongylosis is known based on several human cases and limited field surveys [6]. Our aim of this review is to provide a thorough analysis of the available literature on human angiostrongylosis in the Indian subcontinent and contribute to the understanding of angiostrongylosis in the subcontinent. Data for this study were obtained by searching for the word combination “*Angiostrongylus cantonensis*” AND “*Angiostrongylus*” AND “*cantonensis*” AND “ocular angiostrongyliasis” AND (“India” OR (“Sri” AND “Lanka”) OR “Pakistan” OR “Nepal” OR “Bhutan” OR “ “ Bangladesh” OR “Maldives”) in the Web of Knowledge, Scopus, PubMed, and Google Scholar databases. This initial search revealed 23 relevant reports on angiostrongylosis in the selected geographic area. Then, references in all the studies obtained by the initial search were screened, and those with relevant information were added to the search results. Using this search method, 28 reports were included in our study, yielding 45 cases of angiostrongylosis in humans between 1966 and 2022. In the context of this study, we define the Indian subcontinent as Pakistan, India, Maldives, Nepal, Sri Lanka, Bangladesh, and Bhutan.

## 2. Biology and Life Cycle of *Angiostrongylus cantonensis*

A unique feature of the life cycle of *A. cantonensis* is its low host specificity at the intermediate host and paratenic host levels [1,7]. Natural definitive hosts include rats of the tribe Rattini (as defined by Lecompte et al.) [8]), with three species, *Rattus rattus* (Linnaeus, 1758), *Rattus norvegicus* (Berkenhout, 1769), and *Rattus exulans* (Peale, 1848), being the most reported definitive hosts [9,10]. Rats become infected by ingesting third-stage larvae (L3) in intermediate or paratenic hosts [2,11]. The ingested L3 penetrates the intestinal wall in the small intestine and enters the bloodstream. They then passively traverse the bloodstream and eventually reach the brain. Strong affinity toward the central nervous system (neurotropism) is the fundamental feature of the life cycle. In the central nervous system, L3 go through two molts and reaches the fifth-stage larvae L5 as immature adults in the subarachnoid space about two weeks after infection. Immature adults in typical definitive hosts continue to migrate to the terminal branches of the pulmonary arteries. There they become sexually mature, mate, and females lay eggs. The eggs hatch into first-stage larvae (L1) in the lung parenchyma, which ascend through the trachea and pharynx, then are swallowed into the digestive tract, and finally, the L1 leave the definitive hosts in feces. Snails or slugs many (= intermediate hosts) take up infected L1 from rat feces and these larvae develop into L3 [10,12,13]; typical developmental stages are depicted in Figure 1.

In addition to typical definitive hosts, gut-brain migration occurs in several aberrant hosts, including humans [2,15], domestic [16,17] and wild mammals [18], and birds [19]. However, L3 do not progress in these hosts and frequently cause eosinophilic meningitis accompanied by severe clinical symptoms, with dogs being the most common aberrant host other than humans. Multiple cases of canine angiostrongylosis in Australia show clinical manifestations of neurological deficits, cranial nerve dysfunction, fecal incontinence, hyperesthesia, seizures, ascending tail paresis, depression, diarrhea, and even death. There are also long-term neurological conditions, including tail paresis and hindlimb ataxia [20,21].

To date, the range of identified intermediate hosts of *A. cantonensis* is quite extensive and includes aquatic and terrestrial gastropods from at least 46 families [22]. Intermediate hosts become infected either by ingesting rat feces or by L1 actively penetrating their body wall or respiratory pores [23]. In intermediate hosts, L1 molt to L2 and then to L3, which persists in the tissues of infected mollusks. However, the L3 can leave the intermediate hosts spontaneously, either during its life or, usually, massively after its death. Outside of the host, L3 can survive in water for up to one month [24], with the ability to infect other intermediate hosts, further prolonging the survival of the larvae [25]. In addition to definitive hosts and intermediate hosts, L3 is also the stage that infects paratenic hosts and dead-end hosts and is the key stage of the cycle from an epidemiological perspective. The range of paratenic hosts includes both invertebrates (crustaceans, centipedes) and vertebrates (fish, amphibians, reptiles) [7], with many others likely still unknown.

## 3. Epidemiology and Clinical Manifestations of Human Angiostrongylosis in the Indian Subcontinent

Humans are a typical dead-end aberrant host. Angiostrongylosis caused by *A. cantonensis* was reported in the Indian subcontinent in both clinical forms as eosinophilic meningitis and/or ocular angiostrongylosis [26]. Generally, humans become infected by ingestion of L3 from tissues of intermediate or paratenic host, contaminated water, or vegetables [24]. The larvae penetrate the intestinal wall and, as they migrate, can cause inflammatory reactions in the organs they pass through [2]. *Angiostrongylus* eosinophilic meningitis (AEM) is the typical syndrome that occurs in humans and other aberrant vertebrate hosts [27]. When L3 reach the CNS, the larvae elicit clinical symptoms due to the direct destruction of nervous tissue, the consequent increase in intracranial pressure, and the host inflammatory response (which might get even higher in the event of larval death) [28,29]. The most common signs are severe headache, vomiting, fever, nausea, neck stiffness, paresthesia, hyperesthesia, and visual dystopia [30]. In children, fever, nausea, vomiting, somnolence, and constipation are more common than in adults [2]. AEM may eventually develop into encephalitis (especially in children), followed by coma and death [30].

Ocular angiostrongylosis appears to be more common in Asia than in the rest of *A. cantonensis* distributional range. A recent review [31] shows that most cases occur in a triangle with India to the west, Okinawa to the northeast, and New Guinea to the southeast. Outside this area, only three records are known, namely in South Africa [32], Jamaica [33], and Hawaii [34], all of them published after 2000. In most instances, ocular lesions are caused by a single nematode invading one eye along the optic nerve [26]. The worm is detected mainly in the anterior chamber or vitreous fluid without damaging the retina [31]; however, the subretinal space is also a common site of infection [35]. Patients often report blurred or floated vision [31]. Other series of ocular symptoms, including fundus changes, eye redness and pain, and progressive vision loss, eventually leading to blindness from patients, are also described by Dio et al. [26]. Optic neuritis may develop as a rare complication [31]. Treatment can be delivered in a variety of ways (e.g., surgery, laser, corticosteroids); it usually does not significantly improve visual outcome and focuses on preventing further damage caused by the parasite [35]. Co-occurrence of ocular form and AEM was recorded in 12 of 42 patients, summarized by Feng et al. [31]; only a single such case has been recorded in the Indian subcontinent [36]. In cases of ocular diseases, the nematodes recovered from human eyes were not molecularly characterized, leaving their identification uncertain.

Ocular angiostrongylosis represents an unusually high percentage of human *A. cantonensis* infections in the Indian subcontinent (Table 1), with 13 of 44 (29.5%) published cases from the subcontinent having ocular symptoms. This is even more evident in Sri Lanka, where 5 out of 7 (71.4%) of the ocular cases occur. In comparison, reports from Thailand and China estimate the prevalence of ocular cases to be around 1% [26,35,37]. The difference in the prevalence of ocular angiostrongylosis may suggest that an enormous number of AEM cases remain undiagnosed or unpublished or that *Angiostrongylus* migration within aberrant hosts in the Indian subcontinent differs from those in other parts of the world. The distribution of reports suggests that there may be many undiagnosed or unpublished cases (Figure 2). All published cases are from large metropolitan areas (e.g., Mumbai, Kottayam, Chennai, New Delhi, Colombo) with large hospitals that can diagnose and report the disease to a scientific community. In addition to the geographic pattern, records are also accumulating temporally (e.g., both reports from Madhya Pradesh are from 2019; [38,39], suggesting that the diagnosis of a case attracts attention and sparks eagerness to find more cases. In addition, rat and snail investigations are usually conducted after human cases have been discovered [40,41]. Regarding the age distribution of the patients, the ocular and AEM rate in children seems to be like that in adults [38,42,43,44,45,46,47,48], the small number of pediatric cases does not provide clear evidence. Notably, many case reports associate the infection with the consumption of monitor lizards; however, there are no data on the prevalence of *A. cantonensis* infection in saurian reptiles in the subcontinent (see below).

Accurate and prompt diagnosis of eosinophilic meningitis caused by *Angiostrongylus cantonensis* is critical for appropriate treatment and prevention of sequelae. In general, systemic clinical examination, laboratory tests, and imaging studies can suggest meningitis caused by *A. cantonensis* in endemic areas, however, cannot provide final confirmation of an etiological agent. Elevated count of eosinophils in cerebrospinal fluid (CSF) is an important diagnostic feature and often the first step in the diagnostic process [2,28,49,50].

AEM can be diagnosed by immunodiagnostic methods using purified antigens or monoclonal antibodies in CSF or serum. Numerous techniques, including ELISA, immunoblot assays, gold immunochromatography, and rapid dot immunogold filtration assays, have been used for decades. However, these techniques may have limitations as the antibody may be undetectable in the early stages of infection [3,30,51,52]. Molecular diagnostic methods have been employed as a robust diagnostic tool based on PCR, real-time PCR, LAMP, and recombinase polymerase assay (RPA), which has the potential to improve AEM diagnosis by enabling highly sensitive detection of *A. cantonensis* DNA in a patient’s CSF, serum, or other materials [51,52,53,54,55,56,57,58,59,60,61].

In the Indian subcontinent, 45 previously reported AEM cases were diagnosed based on factors such as a history of eating raw monitor lizard meat, clinical examination, laboratory findings (elevated eosinophils in blood or CSF), ophthalmologic examination and imaging techniques [48,62,63,64,65,66,67], supported by morphological identification of nematodes in cases when they were retrieved. Immunoblotting and qPCR were used to confirm AEM in two cases only [46,63], Table 1.

**Table 1 pathogens-12-00851-t001:** Overview of reports of human angiostrongylosis on the Indian subcontinent, showing clinical form and diagnostic methods involved; definitive and intermediate hosts’ data are included for those studies where they were associated with the investigation of human cases.

Country: State	EM Cases	OC Cases	Diagnostics Method	Monitor Lizard Consumption	References
India: Delhi		1	Slit lamp examination		[43]
India: Delhi		1	Clinical examination, ultrasound B scan, and fundus fluorescein angiography		[42]
India: Karnataka	1		History of raw monitor lizard meat consumption, clinical presentation, and EE in CSF	+	[64]
India: Karnataka	1		Serum and CSF antibodies to *A. cantonensis* 31-kDa antigen positive and magnetic resonance imaging (MRI)	+	[63]
India: Karnataka	1		MRI findings, CSF examination, larvae in CSF wet mount	+	[44]
India: Kerala	5		History of raw monitor lizard meat consumption, clinical presentation, and EE in CSF	+	[66]
India: Kerala	10		History of raw monitor lizard meat consumption, clinical presentation, EE in CSF., larvae in CSF wet mount, MRI	+	[48]
India: Kerala	3		EE in CSF and peripheral blood		[62]
India: Kerala	1 *		Ophthalmological examination, EE in CSF, and microscopic examination of the worm retrieved from eye		[36]
India: Kerala	1		EE in CSF, real-time PCR for *A. cantonensis*		[46]
India: Maharashtra	1		Not mentioned		[6]
India: Maharashtra	2		History of raw slug consumption and EE in CSF		[68]
India: Maharashtra	2		EE in CSF and, peripheral blood	**	[47]
India: Maharashtra	1		Histological examination of brain tissue after autopsy		[65]
India: Madhya Pradesh		1	Slit lamp examination and anterior segment optical coherence topography (AS-OCT)		[38]
India: Madhya Pradesh		1	Slit lamp examination and microscopic examination of the worm retrieved from eye		[39]
India: Puducherry	1		History of raw monitor lizard meat consumption, clinical presentation, EE in CSF analysis	+	[67]
India: Telangana	1		Computerized tomography (CT) scans, parietal craniotomy examination of the worm retrieved from cerebral abscess		[69]
India: West Bengal		1	Microscopic examination of the worm retrieved from eye		[6]
Sri Lanka		1	Ophthalmological examination and microscopic examination of the worm retrieved from eye		[70,71]
Sri Lanka		1	Ophthalmological examination and microscopic examination of the worm retrieved from eye		[72]
Sri Lanka		1	Fundoscopic ophthalmological examination and microscopic examination of the worm retrieved from eye		[73]
Sri Lanka	1		EE in CSF and peripheral blood		[74]
Sri Lanka		1	Ophthalmological examination		[75]
Sri Lanka	1		History of raw monitor lizard meat consumption, clinical presentation, EE in CSF	+	[45]
Sri Lanka		1	Fundoscopic ophthalmological examination and microscopic examination of the worm retrieved from eye		[76]
Nepal: Kathmandu		1	Slit lamp examination, microscopic examination of the worm retrieved from eye		[77]
Nepal		1	Ophthalmological examination and microscopic examination of the worm retrieved from eye		[78]

Clinical form abbreviations and symbol definition: OC—ocular angiostrongylosis; EE—eosinophil examination; EM—eosinophilic meningitis; *—co-occurrence of OC and EM was reported by Baheti et al. [36]. +—Cases with preceding monitor lizard meat consumption. ******—Reported contact with monitor lizard but not its consumption. Note: There is no single direct record of the presence of *Angiostrongylus* sp. in monitor lizard published from the subcontinent. Thus, all human cases associated with monitor meat consumption (Table 1) are only assumptions made from patients’ anamneses.

## 4. Global Distribution and Prevalence of *Angiostrongylus cantonensis* in the Indian Subcontinent

*A. cantonensis* first became known in China in the 1930s and was observed there frequently on different hosts [10,12,79,80]. After the 1950s, numerous studies demonstrated the presence of *A. cantonensis* on various islands in the Pacific [10,12,81,82] and Oceanic regions [13], islands of the Indian Ocean [83]. In the late 1970s, the parasite was discovered in northeastern Africa [84]. In the New World, *A. cantonensis* has been found in some Caribbean islands, namely Cuba, the Dominican Republic, Jamaica, and Puerto Rico [85,86], and in continental Americas [87]. In recent years, it has become apparent that *A. cantonensis* is spreading at an alarming rate. Recent discoveries have been from South America and Brazil [88], the Canary Islands (Spain) [89], Mallorca (Balearic Islands, Spain) [90], Uganda [91], and North America [92,93]. The life cycle of *A. cantonensis* is typically associated with invasive definitive hosts and intermediate hosts, making it a textbook example of a multiple biological invasion. The African giant snail *Lissachatina fulica* (Bowdich, 1822) may have transmitted the parasite to the Pacific basin, where it spread rapidly within pre-existing rat populations [12]. It has been speculated that human activities such as global transport early in World War II may have contributed to the rapid spread of *A. cantonensis* [10,82].

In the Indian subcontinent, the presence of *A. cantonensis* was first detected in Sri Lanka during a survey by Alicata (1965). However, the first reported human case in Sri Lanka most likely dates from 1925 [70,71], i.e., before the parasite was formally described. If the determination is correct, the 1925 study is the first record of nematodes of the genus *Angiostrongylus* in a human host [94]. Later, *A. cantonensis* was also recorded in India [6]. In 1982, the first intermediate host survey took place in the Indian state of Maharashtra and confirmed the presence of *A. cantonensis* throughout India [40]. To date, the parasite has been detected in Sri Lanka, nine Indian states, and Nepal (see Table 1 and Figure 2). Most records (45 of 32 publications) are human clinical cases, supplemented by a few studies on definitive hosts [6,10,40,41,68,95] and intermediate hosts [40,68,96,97]. Paratenic hosts and incidental hosts other than humans have never been studied in the subcontinent.

## 5. *Angiostrongylus cantonensis* in the Definitive Rodent Host

*Angiostrongylus cantonensis* is thought to be largely associated with three invasive species of Rattini: *R. rattus, R. norvegicus,* and *R. exulans*, with local involvement of a few other rodent hosts [58]. The frequency with which infection spreads to other rodent species is largely unknown. An infection of *Sigmodon hispidus* (Say & Ord, 1825), a rodent host in the rather distant family Cricetidae, has been reported in North America [98].

Current knowledge about the distribution of species of the Rattini in the Indian subcontinent is very inconsistent. Most data relate to a few highly adaptable synanthropic species, while most taxa are endemic rodents with a virtually unknown natural history. According to the comprehensive concept of Rattini (as defined by Lecompte et al.) [8], 32 species of 10 genera (*Bandicota*, *Berylmys*, *Chiropodomys*, *Dacnomys*, *Leopoldamys*, *Micromys*, *Nesokia*, *Niviventer*, *Rattus*, *Vandeleuria*) of rats inhabit the Indian subcontinent [99,100,101,102]. The highest rat diversity occurs in the northeast of the subcontinent (e.g., 15 species in West Bengal, Figure 3), where areas overlap with several species from Southeast Asia (including the *A. cantonensis*). This is followed by Sri Lanka, the Western Ghats, and the Andaman and Nicobar Islands (8–9 species each); the diversity there is due to a high degree of local endemism [99]. Although many *A. cantonensis* records are known from the Western Ghats and Sri Lanka (Figure 2, Table 1), the involvement of endemic species in the life cycle of this parasite has never been studied.

Six species of rats inhabiting the Indian subcontinent were confirmed as *A. cantonensis* definitive hosts. Three species (*Bandicota indica*, *R. rattus*, *R. norvegicus*) are reported as hosts in studies directly from the subcontinent (Table 2) [6,10,40,41,83,95,96,103], while three others (*Berylmys bowersi, Niviventer fulvescens, Rattus exulans*) are known hosts in different parts of their distribution range [104]. From an ecological perspective, four of these species (*B. indica* (Bechstein, 1800)*, R. rattus, R. norvegicus, R. exulans*) are synanthropic pests that frequently encounter humans [102]. The other species *B. bowersi* (Anderson, 1879)*,* and *N. fulvescens* (Gray, 1847) [104], avoid human settlements. From the proven definitive hosts, *R. rattus* probably represents a major source of *A. cantonensis* infections and should be investigated; *R. exulans* is of minor importance due to its limited range in the subcontinent; *R. norvegicus* is typically found in large urban areas and seems unlikely to spread infection in rural areas [99]. On the other hand, data are lacking for several other synanthropic species. A total of 437 *Bandicota bengalensis* (Gray & Hardwicke, 1833) were examined by Alicata, Renapurkar et al. [40,83], and Limaye et al. [95], with no single *A. cantonensis* record. According to Agrawal (2000), this species displaces *R. norvegicus* in large urban areas, especially in Kolkata. If there is a difference in host competence between *B. bengalensis* and *R. norvegicus*, this could be the theoretical reason why only one human case is known from Kolkata, compared to Mumbai or Delhi (Table 1). *Rattus tanezumi* (Temminck, 1845) has only recently been separated from *R. rattus* [105,106], so in the case of *R. rattus* records, it cannot be clearly determined which species was examined. From a geographic perspective, *A. cantonensis* in rats was never surveyed in most of the subcontinent. The most conspicuous areas for further study are in the northeast of the subcontinent and associated islands (e.g., Andaman and Nicobar Islands, *R. rattus* was introduced in the Maldives [107]. (Figure 3). In general, the gaps in knowledge about the definitive hosts of *A. cantonensis* in the Indian subcontinent are compelling, considering that *A. cantonensis* is easily diagnosed and mainly associated with rats.

### 5.1. Angiostrongylus cantonensis in Intermediate Hosts

Like most other metastrongylids, the life cycle of *A. cantonensis* invariably involves mollusks as obligate intermediate hosts. However, the nematode can develop in a wide range of gastropods, with extreme variation in prevalence among different populations [10,22,108]. Environmental factors, rat density, and the ecology of specific snail or slug species are likely responsible for the observed differences [22,58,82]. Importantly, *A. cantonensis* exploits both aquatic and terrestrial mollusks, which is one of the reasons for the differences in the local epidemiology of human infections [109,110]. As for gastropods in the Indian subcontinent, Tripathy and Mukhopadhyay (2015) provided a list of the freshwater mollusks of India [111], and Sen et al. summarized the diversity of terrestrial snails in India [112]. Their conclusion that there are 1129 species of terrestrial snails in India alone shows how difficult it is to grasp an enormous diversity of these invertebrates in the Indomalayan region. In addition, there are several smaller studies that list gastropods from different geographic or ecological parts of the subcontinent, such as mangrove mollusks from India [113] or terrestrial snails from Sri Lanka [114]. Many others also attempt to characterize diversity without providing indicative lists [115,116]. Given the low host specificity so far known in *A. cantonensis*, it is easy to imagine that virtually any of these species could play the role of an *A. cantonensis* intermediate host.

In most studies, invasive snail species are considered more important than native fauna due to their ecology and high population density. The spread of *Lissachatina fulica*, one of the most detrimental invasive mollusk species, is commonly referred to as the gateway for the global spread of *A. cantonensis* [117,118]. *Bradybaena similaris* (A. Férussac, 1822)*, Cornu aspersum* (O. F. Müller, 1774)*, Parmarion martensi* (Simroth, 1893)*, Pila* spp., *Pomacea canaliculata* (Lamarck, 1822), and *P. maculata* are associated with *A. cantonensis* in Southeast Asian countries, Australia, and the Caribbean islands [58,119,120,121,122,123,124,125]. Barrat et al. [58] provide a detailed overview of the prevalence and intensity of infection in species where they are known.

*L. fulica* and *P. canaliculata* are described as invasive in the Indian subcontinent [126,127,128], along with *Laevicaulis alte* (Férussac, 1822), *Physa acuta* (Draparnaud, 1805), and several other species [128,129,130,131,132,133]. *L. fulica* is common in almost all states, locally at densities, with negative impacts on agriculture [134]. *P. canaliculata* has invaded various water bodies in the Indian subcontinent [126], *L. alte* is widely reported in India and is known to have negative impacts on native snail species in the area [135]. Although there is no comprehensive study summarizing mollusk invasion across the subcontinent, the online data (www.iNaturalist.org, accessed on 8 November 2022) show a wide occurrence of the major invasive snails and slugs in India.

To date, few studies have addressed *A. cantonensis* in mollusks in the Indian subcontinent. Limaye et al. reported *A. cantonensis* infection in *Macrochlamys indica* (Godwin-Austen, 1883) [95]; the other few studies in the subcontinent [40,68,96,97] focused on a single species, the invasive slug *L. alte* [68].

### 5.2. Snail Consumption

Limited information is available on the scale of edible snail consumption in the Indian subcontinent. Snail consumption is well-known in some parts of India, such as the northeastern region, West Bengal, and other places such as Bihar, Karnataka, Kerala, Madhya Pradesh, and Tamil Nadu [136,137,138,139]. In these regions, snail meat is well known among urbanites and rural tribal communities for its therapeutic and culinary uses [140]. Although the Indian Council of Agricultural Research (ICAR) has supported the introduction of snail farming, there are few snail farms in the country. Instead, snails are collected from the wild rather than being cultivated [141]. Appendix A provides an overview of mollusk consumption. Freshwater snails, *Pila globosa* (Swainson, 1822), *Bellamya bengalensis* (Lamarck, 1822)*, Viviparus viviparus* (Linnaeus, 1758), and several species of terrestrial snails are among the snails reported to be most consumed in many parts of India [140,142,143,144]. Sharma et al. reported a case of angiostrongylosis in humans after consumption of raw slugs *L. alte* [68] but this species has not been mentioned in studies on the consumption of edible mollusks. 

Consumption of raw or insufficiently cooked snails is a common source of human infection in Southeast Asian countries such as China, Taiwan, Thailand, and Hong Kong, including reports of associated clusters of infection [109,145]. However, nematode larvae are sensitive to high temperatures, and even short boiling kills L3 of *A. cantonensis* in infected mollusks [146]. Snails used in reviewed traditional Indian dishes are always prepared by boiling or frying for 5–10 min with various flavors and spices. Technically, following these procedures prevents the presence of live infectious larvae in cooked dishes. Importantly, many recipes recommend soaking the snails in water for 24 h before use. Together with the initial cleaning, this is a critical moment that deserves attention from an epidemiological point of view. The L3 actively escape from snails [147] and can contaminate cooking surfaces and utensils in high numbers [24]. Reportedly, the water from the soaked snails is used as eye drops to treat conjunctivitis as a traditional remedy [148], which may pose an additional risk of angiostrongylosis since larvae may enter the digestive system through the nasolacrimal duct.

## 6. *Angiostrongylus cantonensis* in Paratenic Reptilian and Amphibian Hosts in the Context of Local Consumption in India

### 6.1. Monitor Lizards (Varanidae)

Both amphibians and reptiles have been identified as natural paratenic hosts of *A. cantonensis* [7,149]. In this context, monitor lizards (Varanidae) are most mentioned; several case reports from the Indian subcontinent include a history of monitor lizard consumption preceding AEM symptoms [44,45,48,63,64,66,67]. Although the taxonomy of monitor lizards has not been clearly established [150], four ecologically distinct species inhabit the subcontinent: *Varanus bengalensis* (Daudin, 1802), *V. flavescens* (Hardwicke & Gray, 1827), *V. griseus* (Daudin, 1803), and *V. salvator* (Laurenti, 1768). As potential *A. cantonensis* paratenic hosts, *V. griseus* and *V. flavescens* can be excluded based on their ecology or rarity [151]. In the reports associating *A. cantonensis* infection with monitor lizard meat consumption, species identification was not discussed, and all cases were automatically assigned to the Bengal monitor lizard—*V. bengalensis*. It is the most abundant species throughout the subcontinent [152], a terrestrial animal inhabiting a wide range of habitats from tropical to temperate [153], and a documented paratenic host of *A. cantonensis* in Thailand [149]; the species has been shown to feed on mollusks. The water monitor, *V. salvator*, could be a second potentially important paratenic host. It is a semiaquatic or amphibious species that lives near bodies of water [154] and has been shown to cross the sea between islands. However, it can also live on land and adapt to a variety of food sources, such as human waste when it seasonally visits tourist sites [151]. Because of its ability to disperse to islands, it is also the only monitor species in the Andaman and Nicobar Islands [152]. Cannibalism is known in both monitor species [151], but its role in the life cycle of *A. cantonensis* remains to be investigated. Despite the above, the assumption that consumption of raw monitor meat is a major cause of angiostrongylosis in humans does not seem to be based so much on facts. *V. bengalensis* was found to be a paratenic host harboring L3 of *A. cantonensis* in Thailand [149], and a subsequent study found 100% prevalence at five of four Thai sites [155]. However, these are the only two studies that directly detected *A. cantonensis* in monitor lizards. All subsequent records associate the cases only with the consumption of monitor lizards mentioned by the patients. In the Indian subcontinent, the presence of *A. cantonensis* in amphibians or reptiles has never been directly demonstrated in any study, but it is likely (see, e.g., Anettová et al.) [156]. Of six records in which patients admit to previous consumption of mnitor lizards [44,45,48,63,64,66,67], only one is described in detail, likely ruling out other possible sources of infection [45]. If all these associations were true, it would mean that 47.6% of human cases were caused by the consumption of reptiles—an unusually high number compared to more sporadic cases from SE Asia, e.g., Yang et al. [157]

### 6.2. Snakes

Little attention was paid to the role of snakes in the life cycle of *A. cantonensis* [158]. In India, the consumption of snake meat is not considered a delicacy. However, numerous articles describe snake meat, gallbladder, and skin used in traditional therapies by tribal communities [159,160,161,162,163,164,165,166]. There have been no published cases of neurological or ocular disease caused by *A. cantonensis* linked with the consumption of snake products. Depending on the region and tribe, their products are used in different ways; the most used snake species are listed in Appendix A.

### 6.3. Amphibians

The role of amphibians in the life cycle of *Angiostrongylus cantonensis* deserves more attention. Limited studies have experimentally demonstrated frogs as paratenic hosts [167], and *A. cantonensis* infection has also been detected in free-living frogs [158,168,169]. Reports of angiostrongylosis in humans due to the consumption of frog meat have come from China, Japan, Taiwan, and the United States [7,158,168]. Unlike monitor lizards, frogs are ignored as a source of *A. cantonensis* infection in the Indian subcontinent. In India, amphibians play an important role in cultural traditions, and tribal communities throughout the country rely on amphibians for a variety of uses, including food and traditional medicine [170,171]. However, frog meat is not commonly consumed in India [172]. Due to the trade ban on the export of frog meat and the restriction on the collection of frogs in India [173], they are not widely consumed in the market, yet frog meat continues to be consumed in tribal communities across the country [161,174,175,176]. The most popular way to consume frogs for medicinal purposes is to boil the meat and eat it dry fried or in soup [176]. The *Hoplobatrachus tigerinus* (Daudin, 1802), *Nasikabatrachus sahyadrensis* (Biju & Bossuyt, 2003), and *Euphlyctis cyanophlyctis* (Schneider, 1799) frogs are the most consumed frog species in India [175,176,177]. Appendix A provides examples of frog meat consumption in India.

To our knowledge, frogs have not been investigated as potential hosts for *A. cantonensis* in India, and no cases have been reported associated with the consumption of frog meat, which contrasts with numerous human cases of angiostrongylosis associated with the consumption of raw monitor lizard meat. Unlike frogs, monitor lizards are large animals that are not consumed every day. It is possible that patients affected by angiostrongylosis are likely to remember eating them, associate these two events, and report this to their physician.

Importantly, most studies on the consumption of amphibians and reptiles in the Indian subcontinent report situations in which the meat of these animals is prepared in various ways, including frying, roasting, and boiling. This should be given more attention in the future because nematode larvae cannot survive in cooked animal meat; rather, the infectious L3 from *A. cantonensis* can contaminate kitchenware and cooking utensils and cause infections in humans.

## 7. Conclusions

Despite numerous cases of angiostrongylosis in humans, *A. cantonensis* research in the Indian subcontinent has lagged behind neighboring Southeast Asia and China. Published studies on the possible life cycle invariably refer to infections in humans. In terms of definitive hosts, *A. cantonensis* has been studied locally only in synanthropic rodents, making the potential sylvatic cycle an unexplored area for future research.

Geographically, *A. cantonensis* is well known in southern India and Sri Lanka, with most records coming from large metropolitan areas with large hospitals and dense human populations. Rat species richness appears to play a minor role, as hotspots of diversity do not overlap with the frequency of human cases. The very narrow focus of the published literature and the uneven geographic distribution suggests that *A. cantonensis* infections may be significantly underdiagnosed. The clinical symptoms divide human angiostrongylosis into two syndromes—eosinophilic meningitis and ocular angiostrongylosis. The separation of the two syndromes raises the question of whether they are caused by the same nematode species. Nematode identification was invariably based on nematode morphology, with no DNA sequences available from *A. cantonensis* throughout the subcontinent. Available molecular data [178,179,180,181,182,183,184,185,186] distinguish three related species, namely *A. cantonensis*, *A. malaysiensis*, and *A. mackerrasae* [183]. The possible presence of *A. malaysiensis* or some yet undescribed *Angiostrongylus* species and the hypothetical infection of humans by other metastrongyloid nematodes further complicates the question of the identity of *A. cantonensis* in the Indian subcontinent. Molecular studies that enable proper identification and description of genetic variability of *Angiostrongylus* in the Indian subcontinent are urgently needed, as is awareness and capacity building in the field of *A. cantonensis* diagnostics among medical professionals.

## Figures and Tables

**Figure 1 pathogens-12-00851-f001:**
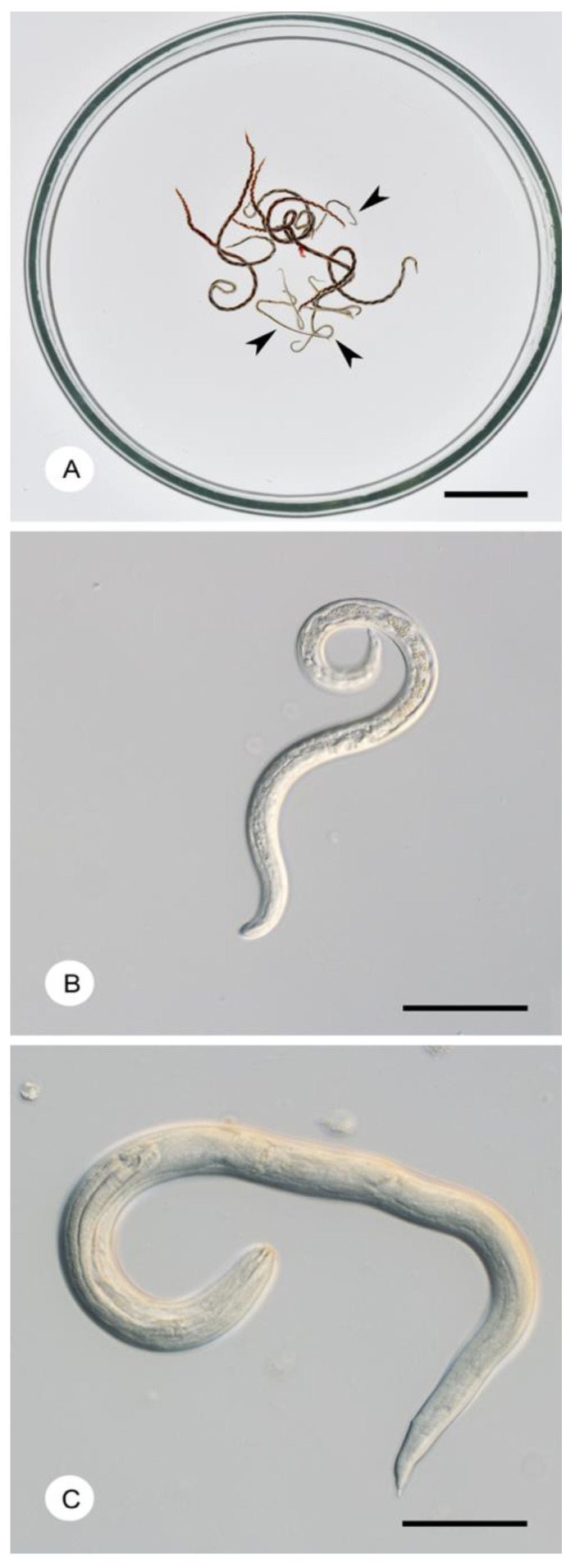
Developmental stages of *Angiostrongylus cantonensis* in intermediate and definitive hosts; Illustrative pictures depict the laboratory strain of *Angiostrongylus cantonensis* from Fatu Hiva, derived from snails from the Marquesas Islands, French Polynesia [14]. (**A**) Mixture of male and female adults removed from pulmonary arteries of a laboratory rat; females are larger (up to 30–35 mm), with typical barber-pole appearance, caused by the interweaving of the intestine and uterus; males (arrowheads) are smaller (max 15–25 mm), whitish, with well-developed bursa copulatrix; scale bar = 5 mm. (**B**) First-stage larva as shed in rat feces, scale bar = 50 µm. (**C**) Third-stage larva from a gastropod intermediate host, scale bar = 50 µm.

**Figure 2 pathogens-12-00851-f002:**
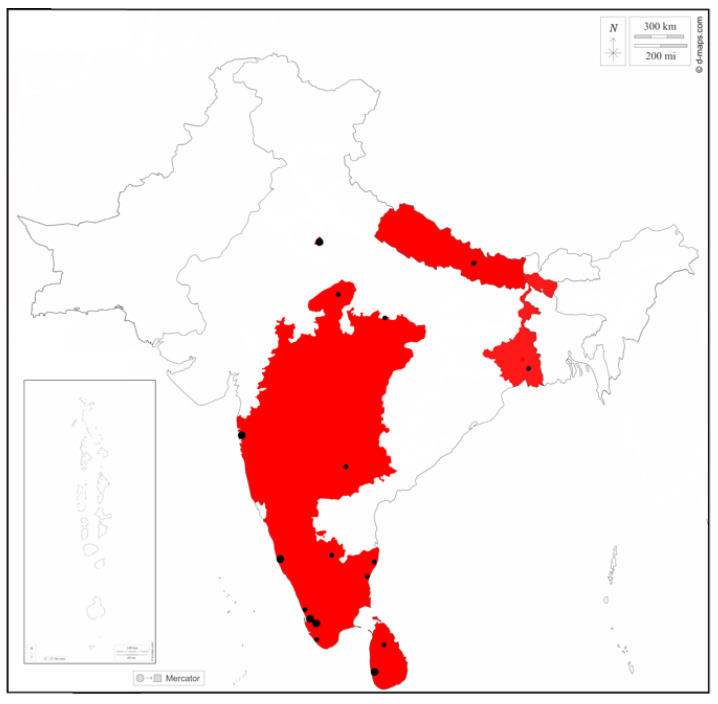
Schematic distribution of *Angiostrongylus cantonensis* reports on the Indian subcontinent. The red areas indicate countries or states with published records of angiostrongylosis, the black dots show cases—large dots multiple cases, and small dots single cases. Exact location of one Nepalese case is unknown, therefore, it is not indicated. The borders between Indian states are not shown, the map background was downloaded from https://d-maps.com, accessed on 8 November 2022.

**Figure 3 pathogens-12-00851-f003:**
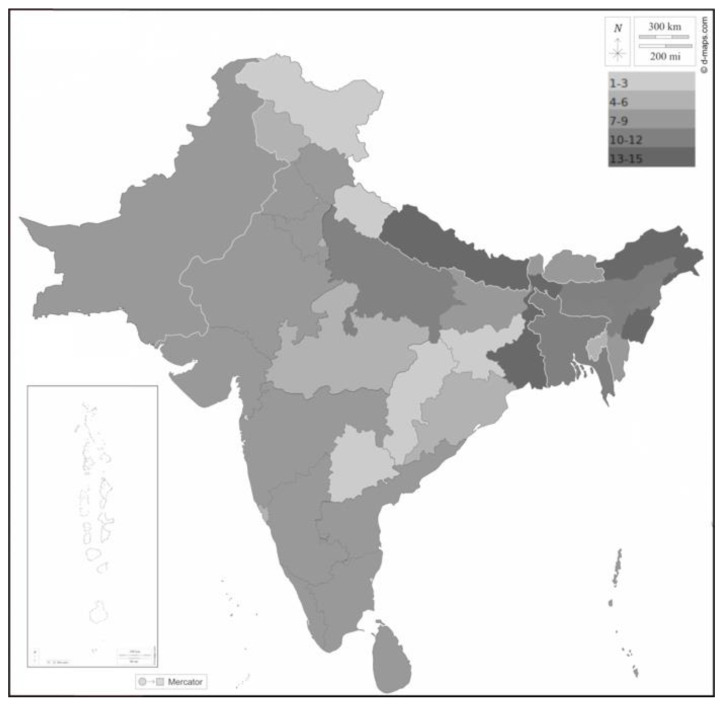
Diversity of rat species (Rattini) on the Indian subcontinent. The grayscale corresponds to the number of rat species described from each country or state, as indicated in the figure. Borders between countries white; coast, borders between Indian states, and outline of the subcontinent black. The map background was downloaded from https://d-maps.com, accessed on 8 November 2022.

**Table 2 pathogens-12-00851-t002:** List of records of *A. cantonensis* from hosts other than humans, as published from the Indian Subcontinent.

Country	State	*Bandicota indica* (Rodentia: Muridae)	*Rattus norvegicus* (Rodentia: Muridae)	*Laevicaulis alte* (Gastropoda: Veronicellidae)	*Macrochlamys indica* (Gastropoda: Ariophantidae)	Reference
India	Kerala	+				[41]
India	Maharashtra		+	+	+	[95,96]
India	Tamil nandu	+				[10,83]
Sri Lanka	Ceylon	+	+			[10,83]

“+”—Definitive and intermediate hosts were investigated in the Indian subcontinent by region.

## Data Availability

Not applicable.

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
