# Peer review of "An Overview of Angiostrongylus cantonensis (Nematoda: Angiostrongylidae), an Emerging Cause of Human Angiostrongylosis on the Indian Subcontinent"

_pathogens, 2023, doi:10.3390/pathogens12060851_

Round 1

Reviewer 1 Report

1)      I recommend consistently using A. cantonensis throughout the manuscript.

2)      Line 68, consider defining the term "neurotropism" for readers who may not be familiar with this specialized characteristic of certain pathogens.

3)      Lines 71-73, provide more details about the clinical symptoms caused by eosinophilic meningitis in aberrant hosts, such as the duration, severity, and potential long-term consequences.

4)      Add more morphological features and distinguishing characteristics of adult and larval stages in Figure 1 to facilitate identification. Although Figure 1A provides a clear picture of a male and female worm, it would be helpful to include additional images or annotations highlighting the differences between the sexes."

5)      Lines 112-120, provide more comprehensive details about the specific studies and surveys mentioned, including the research methods, sample size, key findings, and potential limitations.

6)      Revise Table 1, please move the reference column to the end of the table. The Country and State information is in the first and second columns, respectively, and the Host information is in another column. Additionally, please include separate columns for the occurrences of OC and EM, reported as numbers.

7)      Provide more clarity and detail in Figures 2 and 3 to enhance the visual representation of the data.

8)      Revise Table 2, please list the country in the first column and the state in the second column. Furthermore, please include four separate columns for animal hosts, each reporting the number of occurrences. The last column should contain the reference information.

9)      Clarify the stage found in the paratenic host, such as specifying that line 339 AC refers to the third larval stage of A. cantonensis.

10)   What are the buckets in lines 365 and 366?

Author Response

I would like to sincerely thank you and the other reviewers for the considerable time and effort you dedicated to providing insightful feedback on my work. Your thoughtful comments have been invaluable in shaping the final outcome of the manuscript. Also, I genuinely appreciate the reviewer's informative comments, which highlighted the strengths of my work and pointed out areas that needed improvement. I followed through with the comments. I pointed out the changes that were made to the manuscript. Once again, I would like to express my sincere gratitude for the opportunity to revise my manuscript based on the reviewer's valuable feedback.

1)      I recommend consistently using A. cantonensis throughout the manuscript.

Author response: Corrected A. cantonensis

2)      Line 68, consider defining the term "neurotropism" for readers who may not be familiar with this specialized characteristic of certain pathogens.

Author response:  Explained with a sentence, lines 57-58

3)      Lines 71-73, provide more details about the clinical symptoms caused by eosinophilic meningitis in aberrant hosts, such as the duration, severity, and potential long-term consequences.

Author response: A short paragraph added with an example, lines 97-102

4)      Add more morphological features and distinguishing characteristics of adult and larval stages in Figure 1 to facilitate identification. Although Figure 1A provides a clear picture of a male and female worm, it would be helpful to include additional images or annotations highlighting the differences between the sexes."

Author response:  New version of Fig 1 was prepared, figure legend was expanded: "Males (arrowheads] are smaller, whitish, with clear bursa copulatrix at their caudal end".

5)      Lines 112-120, provide more comprehensive details about the specific studies and surveys mentioned, including the research methods, sample size, key findings, and potential limitations.

Author response:  As this is only an introduction paragraph, it refers to publications that are later described in more detail in the text and tables. A note „Data on the presence in IH and DH are summarised in the following chapters" was added.

6)      Revise Table 1, please move the reference column to the end of the table. The Country and State information is in the first and second columns, respectively, and the Host information is in another column. Additionally, please include separate columns for the occurrences of OC and EM, reported as numbers.

Author response: Changed as required

7)      Provide more clarity and detail in Figures 2 and 3 to enhance the visual representation of the data.

Author response: Changed, though the pdf creation can decrease the resolution again.

8)      Revise Table 2, please list the country in the first column and the state in the second column. Furthermore, please include four separate columns for animal hosts, each reporting the number of occurrences. The last column should contain the reference information.

Author response: as required, new Table 2.

9)      Clarify the stage found in the paratenic host, such as specifying that line 339 AC refers to the third larval stage of A. cantonensis.

Author response: Clarified, line 410

10)   What are the buckets in lines 365 and 366?

Deleted ()

Reviewer 2 Report

The manuscript “An Overview of Angiostrongylus cantonensis (Nematoda: Angiostrongylidae), an Emerging Cause of Human Angiostrongyliasis on the Indian Subcontinent” by Pandian et al. submitted to MDPI (Pathogens) aims at giving an overview of the occurrence of the parasite A. cantonensis in the region, based on reports of human cases found in the literature. Given the crescent importance of this parasite and the pathology associated to human infection, such work is of importance. However, I have issues with the structure and organisation of the manuscript, some of which are highlighted below:

Abstract:

Too long and does not flow well. The authors should observe that there are many species of the genus Angiostrongylus that cause angiostrongyliasis infections, therefore the sentence on line 12-13 is misleading.

If the authors do not change the abstract to give some context on Angiostrongylus malaysiensis it becomes very unclear why the “species identity of nematodes” (Line 36) is included. Except for certain regions of Southeast Asia where co-occurrence of A. cantonensis and A. malaysiensis was proved in both intermediate hosts (snails) and humans infected, the only species of the genus known to cause eosinophilic meningitis in the Indian subcontinent is A. cantonensis.

A. cantonensis – italicized L28

To facilitate the interpretation the use of “monitor” could be replaced by lizard in the abstract.

In general, the abstract does not reflect the manuscript, including the aim described. I could not identified the methodology / inclusion /exclusion criteria used for the selection of papers / reports in the abstract or in the main text.

Introduction:

The authors should choose eosinophilic meningitis or meningoencephalitis and use the same term consistently throughout the text.

Please do not use acronyms for A. cantonensis (i.e., AC) in the text (L49 and so on). This is the parasite of importance in the text and there is no justification for such abbreviation.

Angiostrongylus eosinophilic meningitis (AEM) – which is not used in the text.

Moreover, there is an excessive use of such abbreviations (section 2, L 61-89), IH, PH, DH and CNS for example, which make the reading very confusing. If such terms are not going to be used a lot throughout of the rest of the text, the authors should avoid their use.

Please observe that some references need to be corrected (L64).

Figure 1.  A: I would have removed as it is not possible to visualize any specific structures of the adult parasites that would help others to identify such worms.

In general, there is a lack of references in some sentences and I do not think the order of the sections is the best. Maybe start with the parasite’s biology, life cycle and distribution, followed by a short justification for the study in the Indian subcontinent (including the areas the authors are considering to be it) and the aim. How this systematic review was performed should be described, including inclusion and exclusion criteria for the reports used.

- in the life cycle: please make sure to stress, and even separate, how infection of natural (rodents) and accidental (humans) hosts happens and progress – I found very confusing the way the authors keep changing from one to another in the section 2. Also, information on how human infections are acquired – i.e., by ingestion of L3 infective larvae in raw or undercooked mollusks, paratenic hosts, and contaminated vegetables or water – should be included.

In the next sections the authors might introduce the parasite in the Indian subcontinent, including its intermediate and paratenic hosts in the region, what is known about A. cantonensis human infection in such areas.

Given the conclusion – where the authors mention diagnostic tools – they should add at least a small paragraph explaining the main ways by which the definitive diagnosis of a human eosinophilic meningitis caused by A. cantonensis is achieved.

The table 1 should include how diagnosis of eosinophilic meningitides by A. cantonensis was reached and such results commented in the text.

I would highlight better the link between the reports of the definitive (rodents) and intermediate hosts infections with the risk for human infections in the region studied.

Figure 2. needs definition improved.

Conclusion:

In regards to the A. malaysiensis, the authors only mention it and its potential co-occurrence with A. cantonensis in L 51. Then in the conclusion, the authors go back to this and also mention another species (A. mackerrasae). This needs more context – specially the lack of information on the occurrence of A. malaysiensis in the Indian subcontinent (plus A. mackerrasae seems to be only endemic to Australia). I think the authors can mention the importance of tests that can distinguish species  given the occurrence and of co-infection with different species in specific parts of the world, but given that the course of treatment for such infections are the same, regardless, the most important is to guarantee differential diagnostics and appropriate treatment.

Needs to be improved 

Author Response

I would like to sincerely thank you and the other reviewers for the considerable time and effort you dedicated to providing insightful feedback on my work. Your thoughtful comments have been invaluable in shaping the final outcome of the manuscript. I genuinely appreciate the reviewer's informative comments, which highlighted the strengths of my work and pointed out areas that needed improvement. I followed through with the comments. I pointed out the changes that were made to the manuscript. Once again, I would like to express my sincere gratitude for the opportunity to revise my manuscript based on the reviewer's valuable feedback.

Abstract:

Too long and does not flow well. The authors should observe that there are many species of the genus Angiostrongylus that cause angiostrongyliasis infections, therefore the sentence on line 12-13 is misleading.

If the authors do not change the abstract to give some context on Angiostrongylus malaysiensis it becomes very unclear why the “species identity of nematodes” (Line 36) is included. Except for certain regions of Southeast Asia where co-occurrence of A. cantonensis and A. malaysiensis was proved in both intermediate hosts (snails) and humans infected, the only species of the genus known to cause eosinophilic meningitis in the Indian subcontinent is A. cantonensis.

  1. cantonensis – italicized L28

Author response: Abstract corrected, information on A. malaysiensis is further in the main text body (lines 51-58, 478-480)

To facilitate the interpretation the use of “monitor” could be replaced by lizard in the abstract.

Author response: Changed

Introduction:

The authors should choose eosinophilic meningitis or meningoencephalitis and use the same term consistently throughout the text.

Author response: Corrected to eosinophilic meningitis

Please do not use acronyms for A. cantonensis (i.e., AC) in the text (L49 and so on). This is the parasite of importance in the text and there is no justification for such abbreviation.

Author response: Corrected

Angiostrongylus eosinophilic meningitis (AEM) – which is not used in the text.

Author response: Corrected

Moreover, there is an excessive use of such abbreviations (section 2, L 61-89), IH, PH, DH and CNS for example, which make the reading very confusing. If such terms are not going to be used a lot throughout of the rest of the text, the authors should avoid their use.

Author response: Corrected

Please observe that some references need to be corrected (L64).

Author response: Corrected

Figure 1.  A: I would have removed as it is not possible to visualize any specific structures of the adult parasites that would help others to identify such worms.

Author response: Corrected, new version of Fig 1  was prepared, legend expanded.

In general, there is a lack of references in some sentences, and I do not think the order of the sections is the best. Maybe start with the parasite’s biology, life cycle and distribution, followed by a short justification for the study in the Indian subcontinent (including the areas the authors are considering to be it) and the aim. How this systematic review was performed should be described, including inclusion and exclusion criteria for the reports used.

Author response: Lines 61–71 of the introduction text include more information on  the "method of search".

- in the life cycle: please make sure to stress, and even separate, how infection of natural (rodents) and accidental (humans) hosts happen and progress – I found very confusing the way the authors keep changing from one to another in the section 2. Also, information on how human infections are acquired – i.e., by ingestion of L3 infective larvae in raw or undercooked mollusks, paratenic hosts, and contaminated vegetables or water – should be included.

Author response: Corrected and explained with the continuation of “3. Angiostrongylus cantonensis infections in humans and its occurrence in the Indian subcontinent”.

In the next sections the authors might introduce the parasite in the Indian subcontinent, including its intermediate and paratenic hosts in the region, what is known about A. cantonensis human infection in such areas.

Author response: Explained in the chapter “4. Global invasion and distribution on the Indian subcontinent“.

Given the conclusion – where the authors mention diagnostic tools – they should add at least a small paragraph explaining the main ways by which the definitive diagnosis of a human eosinophilic meningitis caused by A. cantonensis is achieved.

Author response: Explained

The table 1 should include how diagnosis of eosinophilic meningitides by A. cantonensis was reached and such results commented in the text.

Author response: Changed and added with the diagnostic method.

I would highlight better the link between the reports of the definitive (rodents) and intermediate hosts infections with the risk for human infections in the region studied.

Author response: Changed, mentioned with the diagnostic method.

Figure 2. needs definition improved.

Author response: Changed, new version of Fig 2. uploaded

Conclusion:

In regards to the A. malaysiensis, the authors only mention it and its potential co-occurrence with A. cantonensis in L 51. Then in the conclusion, the authors go back to this and also mention another species (A. mackerrasae). This needs more context – specially the lack of information on the occurrence of A. malaysiensis in the Indian subcontinent (plus A. mackerrasae seems to be only endemic to Australia). I think the authors can mention the importance of tests that can distinguish species  given the occurrence and of co-infection with different species in specific parts of the world, but given that the course of treatment for such infections are the same, regardless, the most important is to guarantee differential diagnostics and appropriate treatment.

Author response: conclusion modified

Reviewer 3 Report

This is one of the most interesting articles I've read recently. 

1.     Abstract: Page 1 line 16-17 ‘…our study aims to  investigate the prevalence of angiostrongyliasis in humans…’ – This sentence is not accurate. The authors did not investigate the prevalence of this nematode. The aim of the study is found on p. 2 

 (lines 57-59) and corresponds to the contents of the entire article.

2.     Introduction, Page 1, line 41: ‘Angiostrongylus cantonensis’ – Please add the name of the person who discovered it and the year; this applies to all Latin names of organisms mentioned in the manuscript for the first time.

3.     Page2, line 46-48: Literature references are missing.

4.     -  There is no reference to Figure 1 in the text. A reference to Figure 1 does not appear until p. 5, line 164, but it actually refers to Figure 2.

- Also, please describe where the photos come from; what was the source material? Have they previously been published, and by whom? Did the authors take these photos themselves? If so, please provide detailed information (host, date, location) and a description of the morphology and morphometry of the larvae and other tests performed, as well as the methods used to examine these specimens. What characters were the basis for identifying the species as A. cantonensis?

- Figure 1A is unnecessary and can be deleted, because it does not present any important information.

5.     Page 4, line 108: ‘Lissachatina fulica’ – please add the name of the author who discovered the species and the year.

6.     Table 1 – I suggest making this table horizontal and including the full names of the countries instead of symbols, and making the ‘Reference’ column last instead of first. The data themselves are more important than the work in which they were published.

7.     Page 8, line 309-310 ‘…and can contaminate cooking surfaces and…’ – Is this the authors’ assumption, common knowledge, or information based on the literature? Please clarify in the manuscript.

8.     The conclusion is too long. It should contain a few specific pieces of information summing up what the authors learned from their review of the literature.

Author Response

I would like to sincerely thank you and the other reviewers for the considerable time and effort you dedicated to providing insightful feedback on my work. Your thoughtful comments have been invaluable in shaping the final outcome of the manuscript. I genuinely appreciate the reviewer's informative comments, which highlighted the strengths of my work and pointed out areas that needed improvement. I followed through with the comments. I pointed out the changes that were made to the manuscript. Once again, I would like to express my sincere gratitude for the opportunity to revise my manuscript based on the reviewer's valuable feedback.

  1. Abstract: Page 1 line 16-17 ‘…our study aims to  investigate the prevalence of angiostrongyliasis in humans…’ – This sentence is not accurate. The authors did not investigate the prevalence of this nematode. The aim of the study is found on p. 2 

 (lines 57-59) and corresponds to the contents of the entire article.

Author response: Changed accordingly.

  1. Introduction, Page 1, line 41: ‘Angiostrongylus cantonensis’ – Please add the name of the person who discovered it and the year; this applies to all Latin names of organisms mentioned in the manuscript for the first time.

Author response: Changed for all Latin names.

  1. Page2, line 46-48: Literature references are missing.

Author response: Added

  1. -  There is no reference to Figure 1 in the text. A reference to Figure 1 does not appear until p. 5, line 164, but it actually refers to Figure 2.

- Also, please describe where the photos come from; what was the source material? Have they previously been published, and by whom? Did the authors take these photos themselves? If so, please provide detailed information (host, date, location) and a description of the morphology and morphometry of the larvae and other tests performed, as well as the methods used to examine these specimens. What characters were the basis for identifying the species as A. cantonensis?

- Figure 1A is unnecessary and can be deleted, because it does not present any important information.

Author response:  New version of Fig 1 was prepared, figure legend was expanded, so we prefer to keep this as some of readers might appreciate this general information.

  1. Page 4, line 108: ‘Lissachatina fulica’ – please add the name of the author who discovered the species and the year.

Author response: Added

  1. Table 1 – I suggest making this table horizontal and including the full names of the countries instead of symbols, and making the ‘Reference’ column last instead of first. The data themselves are more important than the work in which they were published.

Author response: Changed

  1. Page 8, line 309-310 ‘…and can contaminate cooking surfaces and…’ – Is this the authors’ assumption, common knowledge, or information based on the literature? Please clarify in the manuscript.

Author response: Based on the published information, a reference added.

  1. The conclusion is too long. It should contain a few specific pieces of information summing up what the authors learned from their review of the literature.

Author response:  conclusion trimmed

Round 2

Reviewer 2 Report

The authors did a good work on revising the manuscript. However, I still think that the Fig1A added is unnecessary and does not add value, unless the features described in the legend are actually visible in the figure.

Author Response

No comments